# Liver-Specific Nanoparticle-Mediated Delivery and MMP-Triggered Release of Veratridine to Effectively Target Metastatic Colorectal Cancer

**DOI:** 10.3390/cancers17193253

**Published:** 2025-10-08

**Authors:** Mahadi Hasan, Morgan Eikanger, Sanam Sane, Krishantha S. K. Wijewardhane, John L. Slunecka, Jessica Freeling, Khosrow Rezvani, Grigoriy Sereda

**Affiliations:** 1Department of Chemistry, University of South Dakota, 414 E Clark Street, Vermillion, SD 57069, USA; mahadi.hasan01@coyotes.usd.edu (M.H.); sajith.wijewardhane@usd.edu (K.S.K.W.); 2Division of Biomedical and Translational Sciences, Sanford School of Medicine, University of South Dakota, 414 E Clark Street, Vermillion, SD 57069, USA; morgan.eikanger@coyotes.usd.edu (M.E.); sanam.sane@usd.edu (S.S.); slune008@umn.edu (J.L.S.); jessica.freeling@usd.edu (J.F.)

**Keywords:** colorectal cancer, veratridine, MMPs, nanoparticles, targeted drug delivery

## Abstract

**Simple Summary:**

Encapsulating an anti-metastatic molecule, veratridine (VTD), by mesoporous silica nanoparticles (MSNs) and calcium carbonate submicroparticles (CCSMPs) enables preferential delivery and release of VTD at the tumor site. The nanoparticles utilize the excessive release of matrix metalloproteinases, specifically MMP-7, by colorectal cancer (CRC) tumors as a “Zip Code” signal. This allows targeted delivery and enrichment of VTD at the tumor site, minimizing systemic exposure.

**Abstract:**

Background: Despite considerable advances to improve colorectal cancer (CRC) survival over the last decade, therapeutic challenges remain due to the rapid metastatic dissemination of primary tumors. This study revealed the apoptotic and anti-growth mechanism of VTD, a previously used anti-hypertensive supplement, can elevate UBXN2A, a known tumor suppressor protein in CRC, and simultaneously enhance intrinsic and extrinsic apoptosis in metastatic cancer cells. Methods and Results: An AOM/DSS mouse model of CRC showed that UBXN2A haplosufficient (UBXN2A +/−) mice treated with VTD had less tumor burden than mice with the full UBXN2A gene treated with vehicle. We have previously shown that casein-coated mesoporous silica nanoparticles (MSNs) offer an effective local delivery of drugs at tumor sites. Our findings demonstrate that the high rate of extracellular release of matrix metalloproteinases (MMPs), particularly MMP-7, by metastatic colon cancer cells, triggers the release of VTD from casein-coated mesoporous MSNs. This shows the “Zip Code” mechanism for the local enrichment of VTD at the tumor sites. After in vitro drug release verification, two independent mouse experiments, a xenograft and a splenolepatic metastatic mouse model of CRC, were used to evaluate the therapeutic efficacy of VTD-loaded and casein-coated carboxylated mesoporous silica nanoparticles, MSN-COOH/VTD/CAS (VTD, 0.2 mg/kg). Animal experiments revealed that MSN-COOH/VTD/CAS (VTD, 0.2 mg/kg) slows down the progress of tumors. Mass spectrometry (MS) revealed improved pharmacokinetics (PK) profile as MSN-COOH/VTD/CAS had less VTD accumulation in non-cancerous organs compared to pure VTD. We further improved nanoparticle targeting and drug release by shifting to calcium-based particles (CBPs). The engineered CBPs demonstrated higher drug-releasing performance. Without the MMPs trigger, MSNs show slow and continuous “drug leak” over longer period of time whereas CCSMPs stops leakage within an hour. Additionally, CBPs showed higher sensitivity to MMP-7 than MMP-9, enhancing the targetability of CBPs for CRC metastatic tumors with excessive extracellular MMP-7. Conclusions: This study introduces a new platform utilizing nanoparticle-based site-specific delivery of a plant-based anti-metastatic molecule, veratridine, with enhanced safety and therapeutic efficacy for the treatment of metastatic CRC.

## 1. Introduction

Colorectal Cancer (CRC) ranks as the second most lethal cancer worldwide because of the high rate of metastasis, as 20% of CRC patients have metastases at initial diagnosis [1]. Dissemination of the primary tumor to distant sites, such as the liver and lungs, is the primary cause of death in the majority of patients [2]. Meanwhile, the rise of CRC in younger adults, commonly diagnosed with the metastatic form of the disease, has dramatically increased over the past decade, showing the pressing need to develop more effective targeted therapies to decrease the high mortality rates associated with metastatic disease [3]. Calculated models based on current trends suggest that the incidence rate of early-onset CRC will increase by 90.0% and 124.2% for patients between 20 and 34 years of age by 2030, respectively [4]. Early-onset CRC is more likely to be diagnosed at metastatic stages of the disease [5], therefore, more effective targeted therapies for metastatic CRC are crucial.

Current evidence indicates that CRC is characterized by the partial suppression of apoptosis, which in turn supports tumors in maintaining their survival, metastasis, and drug resistance features [6]. Understanding the underlying mechanisms of apoptosis in CRC carcinogenesis has provided a new platform for the development of apoptotic inducers, which aim for safer and more therapeutic effects on both primary and metastatic CRC tumors. Additionally, several reports have demonstrated that the combination of an apoptotic inducer and traditional anti-cancer drugs can have a synergistic impact on tumor growth and metastasis, as it has been shown that the induction of apoptosis augments the therapeutic effect of Bevacizumab, an anti-human VEGF antibody, in CRC. We have previously shown that VTD, a plant-based small molecule, can suppress tumor growth by elevating UBXN2A, a tumor suppressor protein with dual targets: (1) the mitochondrial heat shock protein 70 (mortalin) [7,8,9,10] and (2) the Rictor-mTORC2 tumorigenic pathway [11]. Overactivation of mortalin and Rictor-mTORC2 has a dominant inhibitory effect on apoptosis in cancer cells, including CRC [12,13]. A low dose of VTD can significantly suppress tumor growth, colorectal cancer cell migration, and invasion, and induce apoptosis in mouse models of CRC in a UBXN2A-dependent manner.

Designing site-selective and trigger-responsive drug delivery systems is one of the core goals for developing novel solutions to provide safer and more effective targeted anti-cancer strategies [14]. In recent decades, using nano-structured materials as drug delivery carriers has garnered immense attention from chemists [15]. Various drug delivery systems based on organic nanoparticles [16], inorganic nanoparticles [17], carbon-based nano-structures [18], and silica-based nanomaterials [19] have been explored as viable options for achieving the targeted drug delivery and release. After the initial synthesis [20] and application in drug delivery [21], mesoporous silica nanoparticles (MSNs) have gained immense interest from many researchers who have remarkably improved MSN-based drug delivery systems over the past decades. This progress was enabled by the special characteristics of MSNs, such as their mesoporous structure for trapping therapeutic agents and ease of surface functionalization. Furthermore, MSNs have outstanding properties including the following: high chemical and thermal stability, tunable particle size and morphology, uniform and tunable pore size, high specific surface area, large pore volume, unique isolated porous structure, high drug loading capacity, ability for dual surface functionalization (interior and exterior), the gating mechanism of pore opening, as well as high biocompatibility, hemocompatibility and biodegradability [22,23,24]. Due to their versatile surface chemistry, various nanovalves or gatekeeping agents can be attached to the surface of MSNs, which can then be triggered to release cargo, including anti-cancer drugs, by specific stimuli [25,26].

To achieve selective delivery of VTD to the tumor site with MSNs, we decided to utilize one of the features of the CRC metastatic cancer cells, which is mediated through matrix metalloproteinases (MMPs) [27], particularly MMP-7. The catalytically active form of MMP-7 is highly abundant in human colorectal cancer tumors metastasized to their primary metastatic site, the liver [28,29]. While MMPs play a crucial role in tumor invasion and metastasis by degrading the extracellular matrix (ECM) and supporting tumor spread, non-cancerous cells show no elevated release of extracellular MMPs [30,31,32]. Additionally, the activated form of MMP-7 is found exclusively in tumor masses, while it is absent in normal tissues [31]. Meanwhile, in various solid tumors, the level of MMP-7 within tumor tissues is roughly six times higher than in normal cells [33].The elevation of extracellular MMP-7 by metastatic CRC cells is a unique advantage for the selective delivery of drugs to tumor sites, known as the “Zip Code” mechanism. To achieve this selectivity, we utilized proteins, like the gatekeeper molecule casein, which can be chemically hydrolyzed in the extracellular matrix by the Zn-binding site of MMP-7 [34]. This makes MMP-7-triggered gated drug delivery by MSNs nano-cargo profoundly effective against colorectal cancer cells, while MSNs show no release of their cargo in normal tissues. The above mechanism accounts for the effectiveness of casein-coated MSNs loaded with VTD against colorectal cancer cells that we recently reported [32].

We hypothesized that targeted elevation of UBXN2A in cancer cells through site-specific delivery of VTD inhibits tumor development. The proven anti-metastatic activity of VTD combined with MSN-based drug delivery makes it a promising new targeted therapy for patients with metastatic CRC, providing us with a tool to test our therapeutic hypothesis. We designed nanoparticles composed of mesoporous silica and calcium carbonate for the targeted delivery and controlled release of VTD to metastatic cancer cells, where it affects both tumor growth and the expression of UBXN2A. The results reported in this paper demonstrate that the excessive release of MMP-7 and MMP-9 from metastatic colon cancer cells triggers the release of VTD from casein-coated MSNs and calcium carbonate micro- and submicro-particles, supporting our hypothesis and broadening the material platform for developing targeted, gated delivery of anti-cancer therapeutics (Figure 1).

## 2. Materials and Methods

### 2.1. Materials

Cetyltrimethylammonium bromide (CTAB), tetraethyl orthosilicate (TEOS, 99.98%), 3-aminopropyltriethoxysilane (APTES), calcium carbonate (CaCO_3_), sodium hydrogen carbonate (NaHCO_3_), and polyethylene glycol methyl ether (PEGME) were purchased from Sigma-Aldrich (Burlington, MA, USA). Veratridine (VTD, Purity > 98%) was purchased from Sigma-Aldrich (Catalog number #676950-M) or Alomone Labs (Jerusalem, Israel, Catalog number #: V-110). Other chemicals used as received are absolute ethanol, succinic anhydride, 1-ethyl-3-(3-dimethylaminopropyl)carbodiimide (EDC), sulfo-N-hydroxysulfosuccinimide (sulfo-NHS), dimethylformamide (DMF), 2-(N-morpholino) ethanesulfonic acid (MES), glacial acetic acid, blocker casein, and BCA protein assay kit (Thermo Scientific, Waltham, MA, USA), MMP-7 (Millipore-Sigma, Burlington, MA, USA, #M4565), and MMP-9 (Millipore-Sigma, #PF140). All the chemical reagents were of analytical grade.

### 2.2. Instrumentation

A SIGMA FE-SEM (ZEISS Sigma, Oberkochen, Germany) scanning electron microscope and a Tecnai transmission electron microscope (FEI Company, Hillsboro, OR, USA)were used to investigate the morphology of MSNs. For SEM imaging, a 1 mg portion of the synthesized MSN powder was dispersed in 10 mL of absolute ethanol using an ultrasonication probe. A small droplet from the upper part of the dispersion was deposited on the reflective side of the silicon wafer and dried at 60 °C in an oven for 30 min. This specimen was investigated at an accelerated voltage of 2 kV at a working distance of approximately 8.5 mm. For TEM imaging, 1 mg of MSNs was dispersed in 20 mL of nanopure water. One drop (approximately 10 μL) of this mixture was placed on a copper TEM grid, held in place with a tweezer. One to two additional drops were deposited to enhance particle deposition on the grid. The TEM grid was then dried in a desiccator for 48 h, and images were taken at an acceleration voltage of 120 kV. Surface area and pore size distribution were explored by a Quantachrome NOVA-2200e pore size analyzer (Quntachrome Instruments, Boynton Beach, FL, USA) using the N_2_ adsorption method. Brunauer–Emmett–Teller (BET) was employed to confirm the mesoporous nature of MSNs. Pore size distribution was analyzed using the Barrett–Joyner–Halenda (BJH) method from the absorption branch of the isotherm. Before the analysis, 100 mg of the nanoparticle sample was degassed for five hours at 120 °C for 4 h. Surface zeta potential of pristine and functionalized MSNs was measured by a Malvern Zetasizer ZS nano. A 1 mg portion of particles was taken in a glass vial in 10 mL of nanopure water. Then, the sample was sonicated using an ultrasonicator probe for 5 min. Before taking the zeta potential measurement, the dispersions must be sedimented for 1 h. A 1.0 mL portion of a 0.1 mg/mL sample was taken in a folded DTS1070 cell for the measurement. Zeta-potential measurements were taken in five technical replicates and 100 runs for each sample. The amount of drug loading and the percentage of enzyme-triggered drug release were determined using an HPLC system equipped with an Agilent 1100 Series variable wavelength detector (Agilent Technologies, Santa Clara, CA, USA), a SpectraSYSTEM P200 pump (Thermo Fisher Scientific, San Jose, CA, USA), a RHEODYNE 7010 injector (IDEX Corporation, Rohnert Park, CA, USA), and an R-18 reverse-phase 150 mm × 4.6 mm inner diameter column (ThermoScientific, Vilnius, Lithuania). A 60% methanol and 40% ammonium acetate (0.10 M) mixture was used as the mobile phase, which was pumped into the column at 25 °C at a flow rate of 0.35 mL/min after filtration. Data was acquired and analyzed using the software ChemStation (Rev. B. 04. 03 [16]), which is compatible with the Agilent detector. A Thermo-scientific Multiskan SkyHigh Microplate Spectrophotometer was used to conduct the Bicinchoninic acid (BCA) assay to measure the amount of casein coupled with the nanoparticle surface.

### 2.3. MSN Synthesis and Surface Functionalization

MSN synthesis was carried out according to the previously described protocol [35]. Cetyltriethylammonium bromide (CTAB), 5.48 mmole, was completely dissolved in 900 mL of deionized water. Then, 7 mL of 2.00 M sodium hydroxide was added to the solution, and the temperature was raised to 80 °C. After that, 11.47 mL (0.0514 mol) of tetraethyl orthosilicate (TEOS) was added dropwise to the surfactant solution and stirred for 2 h to form a white precipitate, which was collected by centrifugation and subsequently washed (three times with 30 mL absolute ethanol and then three times with 30 mL nanopure water) and separated by centrifuging 11,000 rpm for 8 min on each step. Then, the particles were dried at 60 °C for 12 h to obtain a solid phase that was calcined at 550 °C in air for 5 h, ground by mortar and pestle, and degassed in the oven at 60 °C to obtain the final MCM-41 MSNs (Yield 1.833 g). A 300 mg portion of the synthesized MSNs was taken in 15 mL of absolute ethanol in a round-bottom flask to functionalize the surface with APTES according to a previously described protocol [36]. Briefly, the dispersion was then ultrasonicated for 5 min, and 1 mL of APTES was added dropwise under vigorous magnetic stirring. A 0.6 mL portion of glacial acetic acid was then added to catalyze the reaction, and the mixture was stirred for 24 h. The aminated mesoporous silica nanoparticles (MSN-NH_2_) formed in the mixture were centrifuged and washed with 10 mL of absolute ethanol. Separated particles were dried at 60 °C in the vacuum oven (60 torr) for 24 h. To synthesize carboxylated MSNs (MSN-COOH) from aminated MSNs, the carboxylation process of MSNs was performed according to a known procedure [37]. First, 25 mg of succinic anhydride was dissolved in 25 mL of dimethylformamide (DMF) to form a 1 mg/mL solution. 200 mg of aminated particles were added to the solution and gently stirred for 24 h. Particles were then centrifuged, washed with DMF, and dried in a freeze dryer as a suspension in 1 mL of water to obtain the desired MSN-COOH (175 mg).

### 2.4. Drug Loading, Casein Attachment, and Stimuli-Triggered Drug Release with MSN-COOH

Loading of VTD in MSN-COOH, casein conjugation, and VTD release from the nanocargo were performed according to a previously optimized method [32]. The 0.4 mg/mL stock solution of VTD was prepared by dissolving 4 mg of VTD in 10 mL of 0.1 M PBS (pH 7.4) at stirring for 24 h. A 5 mg portion of synthesized MSN-COOH was added to 1.5 mL of the VTD stock solution and dispersed using an ultrasonication probe for 5 min. The mixture was then vortexed for 5 min and placed in a rotary mixer for 24 h. Next, the mixture was centrifuged at 11,000 rpm for 8 min and washed twice with 1.5 mL of PBS. All the supernatants were collected to calculate the drug loading. After collecting the supernatants, the pellet (containing 5 mg of particles loaded with the drug) was mixed with 1 mL of MES buffer (pH 6.0) using a pipette. A 24 μL portion of 250 mM EDC and 240 μL of 250 mM sulfo-NHS were immediately added to the mixture, which was then gently vortexed and placed in a rotary mixer for 30 min. Then, the particles were centrifuged at 11,000 rpm for 8 min to form a pellet, which was washed with 1 mL of MES buffer, followed by a wash with 1 mL of PBS. After re-dispersing the pellets in 0.4 mL PBS, 1 mL of 0.6% (*w*/*v*) casein was added to the solution and placed in the rotary mixer for 16 h (time optimized for conjugation). The casein-coupled particles were then collected by centrifugation and washed at 11,000 rpm for 8 min with PBS (pH 7.4) buffer. The supernatants were collected for the BCA assay, which was performed according to the previously reported method [38]. In the BCA assay experiment, the working reagent (10.2 mL) was prepared from the BCA assay kit by mixing 10 mL of reagent A (containing BCA in an alkaline environment) with 0.2 mL of reagent B (containing Cu^2+^ ions). A 200 μL portion of the samples was then placed in the wells of the plate reader using PBS 7.4 as a blank. The UV absorbances were measured at l = 562 nm. The solid pellets of the casein-coated VTD-loaded particles (MSN-COOH/VTD/CAS) were then mixed with 1.0 mL of 5 ng/mL and 10 ng/mL of MMP-7 and MMP-9, which were prepared by adding 1X PBS (pH 7.4) buffer to the as-received enzymes. The mixtures were then placed in the rotary mixer and centrifuged simultaneously to obtain data for the drug release profile at set time intervals. Each time, a 300 μL portion of the supernatant was removed for drug release analysis, replaced with an equal volume, and the particles were re-dispersed until the next time interval.

### 2.5. Synthesis of CaCO_3_ Particles, Drug Loading, Skim Milk Coating, and Drug Release

As previously described, calcium carbonate submicroparticles (CCSMPs) were synthesized via a precipitation method [39]. In short, 10.00 g (0.10 mol) of CaCO_3_ was dispersed in 50 mL of nanopure water, to which 12 mL (0.20 mol) of glacial acetic acid was added dropwise to minimize foaming. The mixture was subsequently diluted to a final volume of 100 mL with nanopure water, allowed to stand overnight at room temperature, and then filtered by gravity to obtain the calcium acetate-based solution. To prepare solution A, 20 mL of polyethylene glycol methyl ether (molecular weight = 550) was added to 5 mL of the prepared 1 M calcium acetate-based solution. A 20 mL portion of polyethylene glycol methyl ether (molecular weight = 550) was added to 5 mL of 1M aqueous sodium bicarbonate solution to make solution B. Solutions A and B were combined without stirring and allowed to react for 10 s at room temperature. A cloudy suspension formed, which was then stirred vigorously for 4 h. The mixture was subsequently centrifuged at 11,000 rpm for 10 min to separate the solid product. The resulting white precipitate was washed twice with 40 mL nanopure water, with centrifugation at 11,000 rpm performed after each wash for 10 min. Finally, the product was air-dried, yielding 60 mg of solid material.

The synthesis of CCMPs was carried out using a precipitation method previously reported [39]. 10 mL of a 0.3 M calcium acetate-based solution was prepared by diluting the previously obtained fresh calcium acetate solution with nanopure water. To the prepared solution, 10 mL of a 0.3 M NaHCO_3_ solution was added dropwise at a rate of 5 mL/min under vigorous magnetic stirring (1250 rpm). After the addition was completed, the mixture was left to stand at room temperature for 5 min without stirring, followed by gentle stirring at 150 rpm for 1 h to allow for complete precipitation. The resulting white precipitate was collected by centrifugation (7000 rpm, 5 min), washed three times with nanopure water (2 mL each), and then centrifuged for 5 min after each wash. Finally, the precipitate was redispersed in 1 mL of nanopure water and air-dried at room temperature, yielding 78 mg of solid CCMPs.

VTD loading to calcium carbonate submicroparticles (CCSMPs) and calcium carbonate microparticles (CCMPs) was carried out following a previously established protocol for incorporating eugenol and fluoride into CCMPs [39]. Briefly, 5 mg of particles were dispersed in 1 mL of a VTD solution (0.4 mg/mL in PBS, pH 7.4) and vortexed for 5 min. The solution was then gently mixed using a Roto-Mini rotary mixer (24 rpm) for 24 h. After incubation, the mixture was centrifuged at 8000 rpm for 6 min, and the supernatant was collected. The pellet was resuspended in 1 mL of PBS (pH 7.4), vortexed, and then centrifuged again under the same conditions. Both supernatants were collected, and the concentration of unbound VTD was determined by measuring the absorbance at 220 nm.

For coating of VTD-loaded CCSMPs and CCMPs with casein, 1 mL of skim milk was added to 5 mg of VTD-loaded particles. The suspension was vortexed for 5 min, followed by gentle mixing on a Roto-Mini rotating mixer (24 rpm) for 30 min. The mixture was centrifuged at 8000 rpm for 6 min, and the supernatant was collected. The amount of unbound casein in the supernatant was quantified using the bicinchoninic acid protein assay. Both the BCA assay and drug release experiments were performed following the methods described in Section 2.4.

### 2.6. Cell Culture

We obtained the colon cancer cell lines used in this study from the American Type Culture Collection (ATCC, Manassas, VA, USA). Cell lines were grown in their recommended media, supplemented with 10% fetal bovine serum (Life Technologies, Carlsbad, CA, USA) as well as 100 U·mL^−1^ penicillin and 100 μg·mL^−1^ streptomycin at 37 °C in the presence of 5% CO_2_ except SW48, SW480, and SW620, which were incubated at 37 °C without 5% CO_2_. All examined cell lines were in passages limited to 10 and, per the manufacturer’s protocol, were routinely checked for mycoplasma contamination using the PCR Mycoplasma Detection Kit provided by Applied Biological Materials (ThermoFisher, Waltham, MA, USA).

### 2.7. Western Blot

Cell lysates for WB experiments were prepared using a digitonin lysis buffer [50 mM Tris/HCl, pH 7.5, 150 mM NaCl, 1% digitonin (Sigma-Aldrich) plus 1× mammalian complete protease inhibitor (Research Products International Corp, Prospect, IL, USA). Cell lysates used in WB were normalized for equal loading by NanoDrop using direct absorbance at 280 nM (ThermoFisher Scientific, USA). Samples were loaded onto Novex Tris-Glycine SDS-PAGE protein 4–20% gradient gels. Protein transfer was performed using an iBlot II system to probe the nitrocellulose membranes with the corresponding antibodies. See Appendix A for a detailed list of antibodies used in this experiment.

### 2.8. Crystal Assay and Immunofluorescence Staining

The cytotoxicity test protocol, using a crystal violet staining method, has been previously described. Immunofluorescent assays using BAX and BAD (mouse monoclonal Bax, catalog: sc-20067, and mouse monoclonal Bad, catalog: sc-8044, Santa Cruz Biotechnology, Dallas, TX, USA) were captured with a SP8 confocal on a DMi8 microscope from Leica with high resolution. All images of single cells represent more than 100 cells observed in three independent experiments. Collected signals per cell were measured by ImageJ software and analyzed by GraphPad Prism Version 10.6.0 (890) software.

### 2.9. Human Apoptosis Antibody Array

We used the Human Apoptosis Antibody Array-Membrane (Abcam, Cambridge, MA, USA, #ab134001) to detect the expression of proteins involved in apoptosis that VTD enhances. The array contains 43 apoptotic markers in both the intrinsic and extrinsic apoptosis pathways. The experiments and subsequent analysis were performed according to the manufacturer’s instructions [40]. The UN-SCAN-IT gel analysis software measured the signals recorded on X-ray film using HRP-based chemiluminescence. We used two membrane apoptosis antibody array membranes per treatment, DMSO or VTD, to measure apoptotic markers in the LoVo colon cancer cell line. There were two antibody-fixed spots per protein. The arrays produced substantively similar results.

### 2.10. Assessment of Apoptosis

Colon cancer cells were treated with VTD (100 µM) for 48 and 72 h. A DMSO solvent vehicle was used as a control. Cell apoptosis was assessed using FITC Annexin V (BD Pharmingen, San Diego, CA, USA). Results were collected using a BD Accuri C6 flow cytometer, according to the manufacturer’s instructions and standard protocols (BD Biosciences, Franklin Lakes, NJ, USA). At least ten thousand gated events were recorded per sample. The raw data were analyzed using FlowJo v10.10 software. The results are presented as mean values from three to four independent experiments with treated cancer cells.

### 2.11. Nanoparticle Sterility

To evaluate the sterility of the nanoparticle suspensions, a 50 μL aliquot of either the MSN-COOH/VTD/CAS (VTD, 0.2 mg/kg) or MSN-COOH/EMPTY/CAS (Empty NPs) suspension was aseptically spread onto individual 5% sheep blood agar plates (Hardy Diagnostics, Santa Maria, CA, USA). Plates were then incubated at 37 °C for 24 h under aerobic conditions. Sterility was confirmed by the complete lack of visible microbial growth on the agar surface after the incubation period.

### 2.12. Animals and Preparation

All in vivo experiments received approval from the Institutional Animal Care and Use Committee (IACUC) in accordance with federal guidelines under our approved protocol, 06-05-23-26E. The mice used in this study were housed in an animal facility located within the medical school at the University of South Dakota. Mice were kept on an ad libitum (Ad Lib) feeding schedule with a 12:12 light: dark (LD) cycle. In no case was the maximum IACUC permissible tumor burden exceeded. All surgical experiments were conducted using sterile procedures, and animals were assessed for pain post-surgery before being returned to their cages.

### 2.13. Murine Mouse Model of Colorectal Cancer

We used male and female C57BL/6 mice (Envigo, Denver, CO, USA). The UBXN2A-null mouse line (Ubxn2atm1(KOMP)Mbp) was generated by the Knockout Mouse Project (KOMP, www.KOMP.org accessed on 14 August 2025) on a C57BL/6 background. The colon cancer model in this study involved two chemicals: Azoxymethane (AOM) and Dextran Sodium Sulfate (DSS) (MP Biomedicals LLC, Solon, OH, USA). Mice were first treated with AOM, then given DSS (2% in drinking water) for one week. The AOM/DSS protocol for C57BL/6 mice has been well established as a fast and effective model of CRC. Importantly, this model mimics human CRC with tumors in the lower colon and rectum. IP injections of Veratridine (VTD) or 0.01% EtOH (vehicle) were administered for 5 weeks after AOM injection, every other day (q.o.d). Mice were imaged with ultrasound at three points: before treatment, after treatment, and right before sacrifice. Three-dimensional colon reconstructions were completed for each time point. Image analysis, 3D reconstruction software, and ultrasound contrast optimization were performed as previously described. Necropsy and colon imaging occurred immediately after sacrifice.

### 2.14. Xenograft Mouse Model of Colorectal Cancer

We used eight male and eight female Athymic Nude Foxn1nu mice (Envigo, Denver, CO, USA). 50,000 iRFP-LS-174T cells suspended in 200 mL of Hank’s buffer were injected subcutaneously into the left and right flanks using a 1/2 mL 27Gx3/8 needle. The injection site was wiped with a cotton tip soaked in ethanol. Treatment began 24 h post-cell implantation, with the following strategy q.o.d throughout the experiment: 0.01% EtOH (vehicle), Veratridine (VTD, 0.1 mg/kg), MSN-COOH/VTD/CAS (VTD, 0.2 mg/kg), and MSN-COOH/EMPTY/CAS (Empty NPs). Treatment was administered using retroorbital injections (~50 µL total volume administered per injection), a reproducible venous administration route) for an intravenous drug delivery approach [41]. Tumors were imaged weekly using LI-COR Odyssey CLx (LICORbio, Lincoln, NE, USA) and Leica Vevo 3100 (VisualSonics, Toronto, ON, Canada) ultrasound technology. Image analysis, 3D reconstruction software, and optimization of ultrasound contrast were performed as previously described [42]. Imaging of the resected tumors was performed immediately following animal sacrifice.

### 2.15. Orthotopic Implantation of Cells for a Splenolepatic Mouse Model of Metastatic Colorectal Cancer (CRC)

To generate a stable hepatic metastasis mouse model of murine colorectal cancer, we optimized a CRC metastatic model in which cancer cells are injected into the spleen to develop tumors in the liver, using the human LS174T metastatic colon cancer cell line with Ras and p53 mutations. However, based on the genetic profile of CRC cell lines, the anti-cancer function of VTD-UBXN2A axis can influence the treatment response through different signaling pathways. The advantage of this orthotopic CRC model is that it mimics the human CRC development process, allowing for the study of cancer cell invasion in liver tissue, a natural environment chosen by metastatic CRC cells. Before surgery, mice received 0.015 mg/mL of Buprenorphine, and pain medication was continued twice daily for 2 days after surgery. Mice were anesthetized with isoflurane (4% for induction and 2.5% for maintenance). A small midline incision was made to access the peritoneal cavity and reveal the white mesenteric tissue connected to the spleen, which was gently pulled outside the abdomen onto sterile non-woven gauze sponges (Medline). Once outside the cavity, the spleen was kept moist with a few drops of sterile saline. In the first experiment, eight male and eight female Athymic Nude Foxn1nu mice (Envigo, Denver, CO, USA) were used. Before surgery, 100,000 LS-174T cells suspended in a total volume of 40 μL (20 μL of cells in Hank’s Buffer plus 20 μL of Corning Matrigel Matrix #356224) were injected into the anterior spleen using a 100 μL Hamilton syringe fitted with a 30 G needle. A sterile cotton swab was applied to the injection site to limit leakage, and a few additional drops of sterile saline were used to rinse any cells that may have leaked. The abdominal muscle and skin were closed separately with 6-0 PTFD (Unify^®^) sutures and 7 mm mouse staples, respectively.

Starting 48 h post-surgery, we began treatment with MSN-COOH/VTD/CAS (VTD, 0.2 mg/kg) and MSN-COOH/EMPTY/CAS (Empty NPs). Treatment was administered via Retroorbital injection [41] for intravenous drug delivery q.o.d. for 4 weeks (~50 µL total volume administered per injection). Mice were imaged weekly using LI-COR Odyssey CLx and Leica Vevo 3100 ultrasound technology. To monitor the growth rate of tumors established in the liver, we used the IRDye 800CW RGD Optical Probe (LICORbio, Lincoln, NE, USA, #92609889) according to the manufacturer’s instructions. This BrightSite near-infrared (NIR) fluorescently labeled RGD imaging agent targets the overexpression of integrins on CRC metastatic tumors [43].

### 2.16. Statistical Analysis

All statistical values presented in this study were analyzed using GraphPad Prism 10. The calculation of sample size by power analysis for animal experiments and post in vivo statistical analysis was validated by Dr. Mark Williamson (the Statistician, Biostatistics, Epidemiology, and Research Design DaCCoTA Core, University of North Dakota). Raw data were analyzed for normal distribution and homogeneity of variance to ensure all assumptions were met. Using the recommendations by GraphPad Prism software, we removed outliers before transforming and analyzing the data. Western blot bands captured by LI-COR were quantified using ImageStudio version V software. The UN-SCAN-IT gel analysis software quantified Western blot bands captured by Azure Imager or X-ray films. All biological analyses were performed using a Student’s *t*-test, one-way ANOVA, or multiple-ANOVA with corresponding post hoc tests to determine significance. Bar graphs represent data, and values are expressed as the mean ± standard error of the mean from at least three independent experiments. A *p*-value of 0.05 or less is considered statistically significant.

## 3. Results

### 3.1. Veratridine (VTD), a Novel Small Anti-Cancer Molecule, Induces Apoptosis via Intrinsic and Extrinsic Pathways

Substantial evidence shows that inducing apoptosis can slow or even halt the progression of metastatic CRC. Anti-cancer molecules that activate both intrinsic and extrinsic apoptotic pathways can effectively enhance cancer cell sensitivity to therapy [44,45]. We have demonstrated that VTD induces intrinsic apoptotic pathways in various colon cancer cells [7]. Targeting the extrinsic apoptotic pathway, which causes cell death independently of the p53 tumor suppressor protein, offers an effective therapeutic approach to induce apoptosis in invasive cancer cells. Plant-derived molecules are attractive options for cancer treatment because they can target multiple key components of the apoptotic pathway, thereby activating both intrinsic and extrinsic apoptosis pathways in human cancer cells. Based on the prominent apoptosis-inducing effect of VTD observed in cancer cells [7], we hypothesize that VTD affects both the extrinsic and intrinsic pathways, providing a promising strategy for targeting metastatic cells.

We treated LoVo, an APC/RAS mutant colon adenocarcinoma cell line, with VTD (10 µM and 100 µM) for 48 h. There was a significant increase in Annexin V, an apoptotic marker, in a dose-dependent manner (Appendix A). Using a human apoptosis antibody array that detects 43 human apoptotic proteins involved in both intrinsic and extrinsic pathways, we examined the status of LoVo cells’ extrinsic and intrinsic pathways after treatment with DMSO and VTD (100 µM) (Appendix A). Equal amounts of total lysate protein from DMSO- and VTD-treated LoVo cells were loaded onto the Abcam Apoptosis Antibody Array membrane and exposed to X-ray films. We found an increase of 25% or more in 32 out of 43 apoptotic proteins involved in both pathways in VTD-treated cells compared to controls (Appendix A). Notably, BID proteins increased by 32%, and Caspase 8 increased by 44% in VTD-treated cells (Appendix A). Since Caspase 8 and BID are key components of the extrinsic pathway, this suggests that VTD activates this pathway. Based on apoptosis assays with LoVo, a primary CRC cell line, we repeated the experiment with HCT-116, another primary CRC cell line with high metastatic potential. Western blot analysis showed that HCT-116 cells treated with VTD (100 µM) for 72 h had elevated UBXN2A levels and significant increases in BID protein (Figure 1A–C), as well as cleaved Caspase 8 (Figure 1D,E). We then assessed the status of BAX and BAD, two apoptotic markers downstream of BID. It has been demonstrated that BID activation leads to BAX oligomerization, resulting in the release of cytochrome c. BAD is essential for activating the BAK/BAX pathway. HCT-116 cells treated with DMSO or VTD (30 µM and 100 µM) were stained for BAX (Figure 1F,G) and BAD (Figure 1H,I). Activation of BID by VTD increases the levels of BAX and BAD, which are essential for initiating and progressing apoptosis. Our previous studies revealed that UBXN2A’s ubiquitin-like function allows it to target several substrates involved in apoptosis, including mitochondrial HSP70 (mortalin) and the tumorigenic Rictor-mTORC2 pathway. We hypothesize that increased UBXN2A expression by VTD sustains cell death induction during sub-chronic (10-day) treatment. We used three metastatic colon cancer cell lines: SW48, SW480, and SW620. SW480 (primary tumor) and SW620 (lymph node metastasis) demonstrate distinct drug resistance profiles with cell-specific mechanisms. A crystal violet assay showed that sub-chronic VTD treatment significantly reduced cell viability across all stages of colon cancer cells, primary and metastatic (Appendix A). Furthermore, all three cell lines responded starting at a low VTD concentration (10 μM), with maximal effects observed at 100 μM (Appendix A).

These results indicate that VTD may have a dual function in apoptosis in cancer cells by activating the intrinsic and extrinsic apoptotic pathways in a UBXN2A-dependent and likely in a UBXN2A-independent manner (Appendix A). The results presented in Figure 1 suggest that VTD induces the extrinsic apoptosis pathway in two different human CRC cells (HCT-116 and LoVo), as described for other natural plant products [46]. However, the underlying mechanisms require further investigation in future studies.

### 3.2. Veratridine Suppresses Tumor Progression in an AOM/DSS Mouse Colon Cancer Model in a UBXN2A-Dependent Manner

After confirming that VTD can induce both intrinsic and extrinsic apoptosis, we aimed to assess VTD’s anti-cancer potential using an azoxymethane/dextran sodium sulfate (AOM/DSS)–induced colon cancer mouse model. This method of inducing colon tumors has been shown to closely mimic human colorectal cancer by forming tumors in the lower colon and rectum. Additionally, the AOM CRC model resembles human CRC at several relevant tumorigenic pathways, including the dysregulated AKT/mTORC pathway, which is the main target of the VTD-UBXN2A therapeutic approach.

Following the AOM/DSS protocol, we administered a single injection of AOM in C57BL/6 mice, followed by a week of DSS in their water. We could detect tumors at week 5 during the pre-treatment ultrasound. After this ultrasound, the mice (4 per group, mixed males and females) received intraperitoneal (IP) injections of ethanol (0.01%, control) or VTD (0.1 mg/kg) [47], every other day for 28 days (Figure 2A). A total of 14 injections were administered. The timeline and dosage are an optimized VTD treatment regimen, and there is no neuro- or cardiotoxicity [47]. Intraluminal gel ultrasound allowed us to quantify cancer progression in live animals by measuring tumor volume by 3D reconstruction of the tumors at weeks 5 (pre-treatment), 10 (2 weeks post-treatment), and 12 (endpoint), as previously described by our group [42]. The total tumor growth from ethanol (0.01%) treated wild-type (WT) mice (Appendix A), VTD (0.1 mg/kg) treated heterozygous haploinsufficient UBXN2A (+/−) mice (Figure 2B,C), and VTD (0.1 mg/kg) treated WT mice (Appendix A), was measured at the pre-treatment, post-treatment, and terminal stages. The tumor growth rate was significantly different between UBXN2A (+/−) and WT at the pre-treatment ultrasound stage (Figure 2D). These results can be attributed to the reduced UBXN2A expression in the haplosufficient mice compared to the WT with the full UBXN2A gene at this time point.

At weeks 10 and 12, WT mice treated with VTD (0.1 mg/kg) (Figure 2E,F, red, and Appendix A) had significantly slower tumor progression compared to the ethanol (0.01%, control) group (Figure 2E,F, red versus blue, and Appendix A). More importantly, VTD (0.1 mg/kg) moderately but significantly decreased tumor growth in the UBXN2A (+/−) mice (Figure 2E,F, green versus blue) despite having half the amount of UBXN2A as their WT counterparts. The inhibited tumor growth observed in weeks 10 and 12 in VTD (0.1 mg/kg)- treated UBXN2A (+/−) mice indicates that the elevation of UBXN2A by VTD can overcome pre-existing low UBXN2A levels and actively slow tumor growth. These results suggest that VTD’s ability to inhibit tumor growth is primarily mediated via the UBXN2A-tumor suppressor axis, whose activation is associated with the suppression of multi-tumorigenic pathways in CRC [7,8,10].

### 3.3. Pristine and Functionalized MSNs: Synthesis and Characterization

In vitro and in vivo experiments, as shown in Figure 1 and Figure 2, suggest that VTD has a strong anti-growth and anti-metastatic effect on CRC cells. However, high doses of VTD have been reported to be toxic to both nervous and cardiac systems due to specific interactions with the voltage-gated sodium channel (Nav) [48,49]. We have shown that the concentration of VTD we are using has minimal adverse cardiac or neurological effects in mice. We hypothesized that local delivery of VTD using nanoparticle technology could still increase UBXN2A levels to exert its anti-cancer effect, enabling a high concentration of VTD at the tumor site and a lower overall body exposure. The concentration of VTD encapsulated in the nanoparticles, combined with site-specific drug delivery, would minimize Nav-related adverse effects. The size of the nanoparticle also prevents the drug from crossing the blood–brain barrier and accessing the nervous system. We designed, produced, and tested a mesoporous silica nanoparticle (MSN)-based nano-carrier for targeted delivery of VTD directly to CRC tumor sites, as outlined in Figure 1. In a basic medium, MSNs were synthesized using a sol–gel template method from CTAB and TEOS. To create carboxylated mesoporous silica nanoparticles (MSN-COOH), MSNs were surface-functionalized with APTES, followed by carboxylation with succinic anhydride. These carboxylated MSNs are suitable for conjugating the gatekeeping molecule casein through EDC and sulfo-NHS covalent linking, thereby sealing the pores of the particles to contain VTD until it encounters a metastatic cancer cell producing MMP-7, which triggers drug release. The morphology of bare MSNs was characterized by scanning electron microscopy (SEM) and transmission electron microscopy (TEM). The MSNs are spherical with a particle size of approximately 60–90 nm before surface functionalization (Figure 3A,B). The N_2_ adsorption/desorption isotherm shows a type IV isotherm with H1 hysteresis loops, indicating mesoporosity with inter-particle porosity and well-arranged cylindrical channels (Figure 3C). The Barrett–Joyner–Halenda (BJH) pore size distribution, measured by plotting pore diameter against absorption pore volume, reveals a narrow pore size distribution with a peak around 3.5 nm, confirming their mesoporous nature (Figure 3D). Surface functionalization of MSNs was assessed by monitoring the positive charge from protonated amino groups and the negative charge from deprotonated carboxyl groups on the particle surface. The zeta potential values for MSN, MSN-NH_2_, and MSN-COOH are −18.13, +8.33, and −39.09 mV, respectively (Figure 3E). These results confirm the presence of -NH_2_ and -COOH groups on the surface of the mesoporous silica nanoparticles.

### 3.4. VTD Loading, Casein Coupling, and MMP-7 and 9-Responsive VTD Releasing

After generating MSN-COOH, we loaded VTD into them. The amount of VTD loaded to MSN-COOH was determined by the peak area obtained from High-Performance Liquid Chromatography (HPLC); parameters optimized as flow rate of 0.35 mL/min and pump pressure of 800–900 psi. The VTD peak was observed at a retention time of 18.5 min at a wavelength of 220 nm. A standard curve was constructed using known concentrations of VTD (200, 100, 50, 25, 12.5, and 6.25 μg/mL) to determine the amount of VTD loaded into the nanoparticles (Appendix A). After calculating VTD loading using the calibration curve, the final VTD loading for 1 mg of MSN-COOH was 0.0912 ± 0.0027 mg. The amount of casein chemically coupled to MSN-COOH/VTD was determined by calculating the concentration of non-coupled casein in the supernatants using a standard curve (Appendix A). The calibration curve was constructed for various known concentrations of casein at a wavelength of 562 nm using a UV microplate reader. The absorbance of supernatant samples was measured at the same wavelength to find the concentration of non-coupled casein, thereby determining the amount of coupled casein and the percentage of casein present in COOH/VTD/CAS (VTD, 0.2 mg/kg). The percentage of casein coupled to VTD-loaded nanoparticles was calculated to be 22% for the capping of VTD inside the pores of MSN-COOH. Figure 4 reveals the triggered drug release profiles of MSN-COOH/VTD/CAS nano-cargo at physiological concentrations of MMP-7 and MMP-9. The enzyme solutions (5 and 10 ng/mL of MMP-7 and MMP-9) were introduced to the nano-cargo at time zero, and the release of VTD was monitored over 36 h. In the control experiments with no MMPs, which mimic normal cells, there was 3.5% VTD leakage. The amount of drug released with MMP-7 and MMP-9 at a 5 ng/mL concentration was around 6.5% over 36 h (Figure 4A). Then, we decided to determine the release of VTD at higher concentrations of MMPs that can be released by metastatic cancer cells. Figure 4B demonstrates that 10 ng/mL MMP-7 and MMP-9 lead to approximately 7–8% VTD release from the nanoassembly, respectively. In metastatic CRC cells, the extracellular MMP-7 concentration can reach approximately 24 ng. However, the exact concentration of MMP-7 can vary depending on several factors, including the stage of CRC and individual variations. Each data point demonstrates the average of triplicate experiments with the respective standard deviation. Significant response from MMP-7 or MMP-9 towards the nanocargo, even at 5 or 10 ng/mL concentrations, shows a promising ability of MSN-COOH/VTD/CAS (VTD, 0.2 mg/kg) for site-specific targeted drug delivery and release. The high surface area and tunable pore size of NPs ensure a more robust MMPs-casein interaction, resulting in a higher local concentration of VTD at the tumor site, providing a significant therapeutic impact on cancer cells.

### 3.5. Tumor Suppressive Effect of MMP-Responsive Nanoparticles in a CRC Xenograft Model

After validating the successful and selective release of VTD from MSN-COOH/VTD/CAS nanoparticles triggered by MMP-7 and MMP-9, and our previously published data [32], we wanted to evaluate MSN-COOH/VTD/CAS’s anti-cancer effect in vivo. A xenograft mouse model can reproduce the outcomes of anticancer therapies in patients, providing valuable insight into the sensitivity and efficacy of the tumor to novel therapies. Therefore, we decided to determine the therapeutic efficacy of MSN-COOH/VTD/CAS (VTD, 0.2 mg/kg) in the xenograft CRC mouse model [50].

As previously reported, metastatic CRC tumors predominantly progress in later stages of the disease and therefore have more extracellular release of MMP-7 and MMP-9 at the metastatic site [27,28]. To evaluate the MSN-COOH/VTD/CAS’s effect on tumorigenesis, we injected 50,000 iRFP-LS174T cells subcutaneously into eight male and eight female Hsd: Athymic Nude-Foxn1nu (Envigo) mice to establish a CRC xenograft model. One day after cell implantation, mice began receiving retroorbital injections of EtOH (0.01%, vehicle), VTD (0.1 mg/kg), MSN-COOH/VTD/CAS (VTD, 0.2 mg/kg), and MSN-COOH/EMPTY/CAS (Empty NPs) q.o.d. for 4 weeks (Appendix A). Before the first treatment, we ensured that both nanoparticles, with and without VTD, had a stable, homogenized phase suitable for IV injection (Appendix A). The sterilization process of nanoparticles (see Section 2) eliminates any local and systemic contamination, making them suitable for repeated IV injection (Appendix A). Every week, mice were imaged using the LICOR Odyssey to measure the iRFP-fluorescent signal on the LS174 cells, allowing us to evaluate tumor progression in live mice. We utilized two mice per group with two growing tumors on each side of the mice’s flanks. This approach enabled us to have four individual tumors (n = 4) per treatment for both male and female cohorts, minimizing the number of experimental animals while maintaining acceptable statistical power. The level of the iRFP-signal intensity per tumor in live animals is used for the tumor growth rate. The statistical analysis of tumor growth per group revealed that tumors developed in male mice receiving control (0.01% EtOH) or MSN-COOH/EMPTY/CAS were larger than those in mice receiving VTD alone or MSN-COOH/VTD/CAS (VTD, 0.2 mg/kg) at week 4 (Figure 5A,B and Appendix A). In contrast, the treatment groups in female mice showed no significant difference, indicating that the LS174 cell-based tumors in female mice are less sensitive to VTD’s anti-growth effects (Figure 5C,D and Appendix A). Mass spectrometry assessed VTD (pg/mg) accumulation in the serum, brain, heart, liver, and tumor (Appendix A). There was no VTD measured in the serum which supports previous reports about MMPs low activity in the serum [33]. The results revealed less accumulation of VTD in the heart and brain in both male and female mice treated with MSN-COOH/VTD/CAS (VTD, 0.2 mg/kg) compared to pure-VTD. Interestingly, we observed an enrichment of VTD in the liver tissue in MSN-COOH/VTD/CAS (VTD, 0.2 mg/kg) treated mice. Due to their size, the nanoparticles are trapped and VTD is metabolized in the liver, making MSNs a potential delivery strategy for metastatic CRC developed in the liver (Appendix A). Measurement of VTD in the tumor showed accumulation of VTD in male mice treated with pure VTD and MSN-COOH/VTD/CAS (VTD, 0.2 mg/kg). Female mice showed no detectable levels of VTD, which can be attributed to fast metabolism of VTD or a technical limitation. Western blot in female mice showed no meaningful changes in UBXN2A and pAKT473 protein in female mice (Appendix A). There is a remarkable elevation of UBXN2A in male mice treated with MSN-COOH/VTD/CAS compared to pure VTD (Appendix A). The absence of elevated UBXN2A levels in the mice treated with pure VTD may be due to the significant necrotic tissue in the tumor after 4 weeks of treatment, as previously reported by our group. Expectedly, we observed a partial reduction in pAKT473 protein in male mice treated with MSN-COOH/VTD/CAS after normalization. 

As previously described [51], for similar plant-based anti-cancer molecules, both pure VTD and MSN-COOH/VTD/CAS (VTD, 0.2 mg/kg) show reduced tumor growth in xenograft tumors, particularly in male mice. The delivery of VTD encapsulated in MSNs reduces the exposure of non-cancerous tissue to the drug and simultaneously allows local enrichment of VTD at the tumor site within the liver tissues, the primary site of metastasis in CRC. This finding demonstrates MSNs can be a promising delivery tool for treating metastatic CRC.

### 3.6. MSN-COOH/VTD/CAS (VTD, 0.2 mg/kg) Target and Suppress Metastatic Tumors in an Orthotopic Splenolepatic Mouse Model of CRC

Enrichment of VTD in xenograft mouse livers suggest that MSNs encapsulating VTD could be an effective approach to target metastatic CRC tumors developed in the liver. Therefore, in the next set of experiments, we investigated the effect of VTD and MSN-COOH/VTD/CAS on tumor growth using a modified splenolepatic metastasis model of CRC [50,52]. The portal system allows for the easy spread of metastatic cells from splenic tumors to the liver. We used LS-174T cells to determine whether the development of hepatic tumors in the liver and whether our optimized method will mimic the human CRC metastatic tumors in the liver. Appendix A shows the formation of intrasplenic tumors followed by the appearance of the intrahepatic individual tumor masses. Appendix A illustrates the formation of individual nodules, which represent colonies of cancer cells that have migrated from the primary intrasplenic tumors, providing strong support for the metastatic origin of these tumors.

As previously reported [53], functionalized mesoporous silica nanoparticles (MSN-COOH) of 70–105 nm size will be preferably captured by the liver’s microphysiology system (MPS) and ideally accumulate in the liver, where the metastatic nodules develop. Interestingly, carboxylation of MSNs using succinic anhydride followed by amination with aminating agent 3-aminopropyltriethoxysilane (APTES) increases the particle size by around 10 to 15 nm to be more efficiently captured in the MPS [32]. To verify the dominant accumulation of MSN-COOH in the liver tissues, IRDye 800CW NHS Ester (green) was conjugated to casein-coated nanoparticles and injected into mice to visualize the NPs on the metastatic tumors. Appendix A illustrates the presence of tumor masses in the spleen and liver. LI-COR visualization of IRDye 800CW NHS Ester revealed dominant accumulation of the fluorescent, green-tagged NP in the liver tissues surrounding tumors. The NPs leave the liver tissue via the hepatobiliary excretion mechanism. The low accumulation of green-tagged NPs in the spleen tissue further confirmed the enrichment of NPs in the liver tissues, where metastatic tumors are growing with their enrichment of MMP-7 in the extracellular space.

Following optimization and validation of the splenolepatic metastasis model, 100,000 LS-174T cells were injected into the spleens of athymic nude mice, and tumor growth was monitored for 5 weeks (Figure 6A–C). Treatment with MSN-COOH/VTD/CAS (VTD, 0.2 mg/kg) or MSN-COOH/EMPTY/CAS (Empty NPs) began 48 h after surgery. Before each weekly LI-COR scan, we injected IRDye 800CW RGD Optical Probe, which is designed to target the overexpression of integrins on tumors, to visualize the tumors in the liver. Mice that received MSN-COOH/EMPTY/CAS showed more fluorescence intensity and a higher tumor burden in the liver than those that received MSN-COOH/VTD/CAS (VTD, 0.2 mg/kg) (Figure 6D–G). After necropsy, the resected livers were imaged and weighed. Male mice treated with MSN-COOH/VTD/CAS (VTD, 0.2 mg/kg) had average liver weights of 2.1 g, whereas those treated with MSN-COOH/EMPTY/CAS averaged 5.6 g, due to the larger tumor burden. Female mice showed similar results, with MSN-COOH/VTD/CAS (VTD, 0.2 mg/kg) livers averaging 1.9 g and MSN-COOH/EMPTY/CAS livers averaging 3.7 g (Figure 6H). Since the liver weight of an 8-week-old C57BL/6 mouse without tumors is approximately 1.6 g, the average liver weight of mice treated with MSN-COOH/VTD/CAS (VTD, 0.2 mg/kg) (~2.1 g) indicates that the livers resemble a moderately tumor-free liver at approximately 12 weeks old. Additionally, we counted the number of tumors in the liver for both male and female mice, and MSN-COOH/VTD/CAS (VTD, 0.2 mg/kg) significantly reduced hepatic metastasis and the number of metastatic nodules (Figure 6I). Both the xenograft (Figure 5) and splenolepatic metastasis model (Figure 6) suggest that the MSN-COOH/VTD/CAS (VTD, 0.2 mg/kg) has the potential to become a safer and more effective treatment strategy for metastatic CRC.

### 3.7. Calcium Carbonate Microparticles (CCMPs) and Calcium Carbonate Submicroparticles (CCSMPs): Improved Carriers for VTD Delivery

Based on in vivo results (Figure 5 and Figure 6), we hypothesized that improving the nanoparticle composition could enhance both the release and tumor-targeting ability of NP encapsulated with VTD. Therefore, we examined the potential to expand the MMP-7-triggered release of VTD from calcium carbonate particles, which are expected to have lower toxicity toward normal tissue and faster clearance from the body.

Calcium carbonate submicroparticles were synthesized from calcium acetate, polyethylene glycol methyl ether, and sodium hydrogen carbonate. The 400–800 nm-sized particles of CCSMPs with minimal aggregation exhibited a uniform spherical shape with a porous surface, making them suitable as drug delivery carriers (Figure 7A). Additionally, larger calcium carbonate microparticles were prepared from calcium acetate and sodium bicarbonate using the precipitation method. Scanning electron microscopy shows a uniform spherical shape of CCMPs with particle sizes ranging from 1–2 μm and a porous surface (Figure 7B).

The VTD loading efficiency of CCSMPs and CCMPs in PBS was determined to be 0.0130 ± 0.0018 mg and 0.0118 ± 0.0004 mg of VTD per 1 mg of CCSMPs and CCMPs, respectively. While this result confirms successful VTD incorporation, the VTD loading efficiency is slightly lower in CCMPs than CCSMPs. Drug loading in CCMPs and CCSMPs shows a lower value compared to the drug loading (0.0912 ± 0.0027) efficiency in MSNs. We utilized skim milk as a natural source of casein for its conjugation to CCSMPs and CCMPs. Casein accounts for the majority of milk proteins, making up about 80% of the total protein content. Whole milk contains roughly 3.3% protein by weight, with casein contributing approximately 2.6–2.7%. Although the fat is removed during the skimming process, skim milk maintains a protein profile comparable to that of whole milk. The casein concentration in skim milk typically remains around 2.7–2.8%, as fat removal minimally affects protein levels [54]. Casein conjugation was determined to be 3.41 ± 0.04 mg and 2.37 ± 0.04 mg per 1 mg of CCSMPs and CCMPs, respectively, by the BCA assay. These results indicate a high level of protein attachment, confirming effective casein coating on both CCSMP and CCMP surfaces. The release behavior of VTD from CCSMP-VTD-SM and CCMP-VTD-SM was examined under both enzymatic and non-enzymatic conditions to mimic tumor environments. As shown in Figure 8A, CCSMP-VTD-SM exposed to 5 ng/mL MMP-7 (Figure 8A, blue curve) exhibited a markedly higher cumulative release, reaching approximately 27% within 12 h, and showed no release after that point. Upon 5 ng/mL MMP-9 (Figure 8A, black curve) activity, 10% VTD was released from the cargo within 6 h before completely stopping the release. As depicted in Figure 8B, CCMP-VTD-SM treated with 5 ng/mL MMP-9 (Figure 8B, red curve) showed moderate release, achieving around 12% until 12 h before the release was stopped, whereas MMP-7-triggered release was found 11% and the release ended after 2 h (Figure 8B, black curve). CCSMP-VTD-SM (Figure 8A, green curve) and CCMP-VTD-SM (Figure 8B, purple curve) systems maintained under non-enzymatic conditions showed negligible release, with approximately 5% and 2% of the total release, respectively, confirming particle stability in the absence of MMPs. Together, these results highlight the enzyme-responsiveness of the calcium-based systems and their potential utility for targeted therapeutic delivery in MMP-rich cancerous environments. It is worth noting that the particle size of CCSMPs (400–800 nm) and their higher sensitivity to MMP-7 enable CCSMPs to be an ideal vehicle to deliver cargo more effectively to the CRC metastatic tumor site while inhibiting their perfusion to the brain via the blood–brain barrier (BBB). Compared to conventional MSN systems, CCSMPs and CCMPs showed different release behaviors in response to MMP treatment. As shown in Figure 4, MSNs released around 7–8% of the loaded VTD over 36 h. Although the overall release was low, MSNs continued releasing steadily over the whole time period compared to CCSCMs and CCMPs, suggesting they may be better suited for slow, sustained drug delivery applications. On the other hand, CCSMPs showed a faster and stronger release, especially with MMP-7, reaching approximately 27% cumulative release within 12 h. CCMPs mainly remained stable, with less than 15% release even under enzymatic conditions. When no enzymes were present, the three systems observed notable differences in drug retention. MSNs showed slow but continuous leakage of VTD over time, reaching nearly 3–4% cumulative release after 36 h. In contrast, both CCMPs and CCSMPs demonstrated lower release, 2.3% and 5.4%, respectively, under non-enzymatic conditions, showing no leak after 1 h. Finally, the in vivo experiments with calcium-based nanoparticles will serve as an ideal model, as the CCSMP release rate can be controlled by the acidic environment of metastatic tumors.

## 4. Discussion

Patients with CRC have a dismal survival rate of 14%, and this is because there are limited therapies capable of safely and effectively targeting metastatic disease [1,2,3]. Our group has and is currently studying the mechanism of the UBXN2A-Rictor-mTORC2 axis [11] and the UBXN2A-mitochondrial HSP70 protein (Mortalin) axis [7,10]. CRC tumors manipulate and change these pathways to their advantage, including evading apoptosis to promote tumor progression and metastasis. Results in this report indicate that VTD can induce both intrinsic and extrinsic apoptosis in CRC. This dual apoptotic function can be attributed to VTD’s dual inhibitory function, which targets the mitochondrial protein quality control (MPQC) machinery in cancer cells with oncogenic mitochondria by targeting and degrading mortalin, a key player in mitochondrial protein quality control [7,10]. Simultaneously or sequentially, VTD-UBXN2A targets the Rictor protein, a key member of the mTORC2 tumorigenic pathway involved in an anti-apoptotic role in tumor tissues [44]. Targeting both apoptotic pathways (intrinsic and extrinsic) induces a more effective apoptosis in both primary and metastatic CRC cells. The AOM/DSS model revealed that pure VTD can potently decrease the growth rate of tumors growing in the lower portion of the colon and rectum in a UBXN2A-dependent manner. Understanding the UBXN2A-independent anti-cancer function of VTD is an ongoing project in our group.

Natural plant-based molecules have been valuable resources for discovering and developing novel biologically active anti-cancer compounds, owing to their unique structures and mechanisms of action [55,56,57,58]. However, most of these natural anti-cancer molecules often react nonspecifically with multiple biological targets rather than specifically affecting a single desired target in pre-clinical models. High concentration of VTD and its binding to sodium channels can induce neurotoxicity and cardiotoxicity [48,49]. We have already shown that VTD at its therapeutic concentration has no toxicity on both the nervous system and cardiac function [47]. Nevertheless, to eliminate the potential toxicity of VTD in normal organs, we have developed a nanoparticle encapsulated with VTD, which is capable of responding to a specific protease released by CRC metastatic cells (MMPs), resulting in reduced systemic exposure of normal organs to high doses of VTD. Figure 3 and Figure 4 show the successful composition of MSN-COOH/VTD/CAS nano-cargo and its VTD-release profile in response to the excessive concentrations of MMP-7 and MMP-9 released by metastatic tumors.

Following the successful performance of MSN-COOH/VTD/CAS (VTD, 0.2 mg/kg) in vitro, we examined the anti-growth function of MSN-COOH/VTD/CAS (VTD, 0.2 mg/kg) in two different mouse models of CRC. Xenograft models of cancer accurately reflect the tumor’s sensitivity and response to novel therapies [50], and our splenolepatic metastatic model of CRC can better recapitulate the tumor microenvironment (TME) of metastatic CRC. In both animal studies, we quantified and tracked tumor growth in live animals, enabling us to evaluate the response and tumor-targeting ability of VTD and MSNs encapsulating VTD. As expected, VTD (0.1 mg/kg) and MSN-COOH/VTD/CAS (VTD, 0.2 mg/kg) treated mice, particularly male mice, had less tumor growth in both xenograft and metastatic tumors in the liver compared to the tumors in mice that received the control. Importantly, MSN-COOH/VTD/CAS (VTD, 0.2 mg/kg) was able to significantly slow the size and number of metastatic lesions in the liver. In previous studies, we have seen that males have a better anti-cancer response to VTD than females. We will consider the sex-specific differences when moving forward in this study, mainly because combination therapy and re-sensitizing resistant cancer is another therapeutic approach to treat metastatic cancers, particularly with the new generation of immunotherapies [59,60,61].

After completing the animal experiments with mesoporous silica nanoparticles, we determined that their targeting and release capabilities could be further enhanced. The MSN-COOH/VTD/CAS (VTD, 0.2 mg/kg) was effective; however, we sought to improve it further. Therefore, we decided to employ another particle for a similar, but improved approach. MMP-7-triggered CCSMPs-based release (around 27%) is 3 times better than MSNs-based release (around 7–8%). Therefore, highly sensitive CCSMPs may offer very different pharmacokinetics, extending our tool set for fine-tuning the timeline of the targeted VTD delivery. The application of calcium carbonate-based particles goes beyond treating cancer since they can be easily digested in a lower pH (5.0) environment [39]. So, as opposed to MSN, CCSMPs may release VTD under low pH in addition to the enzyme-triggered release. This dual trigger release is significant because VTD reduces the expression of MMP-7, mediated through the inhibition of the mTORC2 pathway by metastatic cancerous cells, which would otherwise reduce the effectiveness of a single trigger releasing mechanism.

The next step in this project is to utilize our optimized xenograft and splenolepatic metastatic model of CRC, which incorporates calcium-based submicroparticles, to achieve a higher release rate in response to MMP-7 and MMP-9 with the same selectivity for the tumor site.

## 5. Conclusions

A dual pathway targeting strategy, which targets multiple tumorigenesis pathways simultaneously, can be a more effective strategy for controlling tumor growth than targeting a single tumorigenic pathway. The dual pathway strategy provides more effective therapy and improves the survival rate. Small molecules that simultaneously target multiple tumorigenic pathways can overcome resistance mechanisms that tumors develop to targeted therapies and potentially achieve more significant tumor suppression. Plant-based anti-cancer molecules and their unique structural features can target diverse tumorigenic pathways in cancer cells and simultaneously decrease the development of drug resistance and tumor recurrence in patients. At the same time, combination therapy with plant alkaloids can elevate the therapeutic impact of traditional chemotherapeutic drugs. This shows the novel use of VTD and its ability to target multiple tumorigenic pathways, making it a promising anti-cancer agent. Encapsulating VTD inside nanoparticles can add another level of precision and targeted therapy for patients with metastatic CRC by significantly eliminating VTD exposure to healthy organs.

In this study, we verified that the anti-growth and apoptotic induction effects of VTD can simultaneously or sequentially inhibit primary and metastatic tumor growth in both in vitro and in vivo settings. Additionally, we developed a SMART delivery tool capable of delivering and releasing high concentrations of VTD next to metastatic tumors grown in the liver, minimizing the exposure of VTD to other normal organs. Cell culture, in vivo pharmacokinetic/pharmacodynamic, safety, and therapeutic efficacy studies of a next-generation calcium-based nanoparticle-based delivery system in the CRC metastatic mouse model will define the primary therapeutic platform for the clinical trial.

## 6. Patents

Intellectual property: This long-term product development strategy is protected by an active U.S. patent owned by the South Dakota Board of Regents (Rezvani, K.; Sereda, G. Methods and Compositions for the Treatment of Cancer. U.S. Patent No. 11,717,573 B2, 8 August 2023) and a continuation granted by the U.S. Patent Office: Rezvani, K.; Sereda, G. Methods and Compositions for the Treatment of Cancer. US Patent No. 12,233,129 B2, 25 February 2025.

## Data Availability

The raw data generated in this study will be stored in a secure file managed by the University of South Dakota (USD). The collected data and corresponding final analyzed data from this project will be made available to researchers and analysts without cost. User registration, pre-coordination with Khosrow Rezvani, and pre-approval by the Office of Research at USD are required to access or download files.

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
