# Peer review of "Liver-Specific Nanoparticle-Mediated Delivery and MMP-Triggered Release of Veratridine to Effectively Target Metastatic Colorectal Cancer"

_cancers, 2025, doi:10.3390/cancers17193253_

Round 1
Reviewer 1 Report
Comments and Suggestions for Authors
The manuscript presents a novel nanoparticle-based strategy for targeted delivery of veratridine (VTD) to metastatic colorectal cancer (CRC), exploiting the overexpression of MMP-7/MMP-9 in tumor microenvironments. The authors demonstrate that casein-coated mesoporous silica nanoparticles (MSNs) and calcium carbonate submicroparticles (CCSMPs) enable MMP-responsive VTD release, leading to significant anti-tumor effects in both xenograft and spleno-hepatic metastasis models. The study also explores the mechanistic role of UBXN2A in mediating VTD-induced apoptosis.
The work is innovative and clinically relevant, combining nanomedicine, targeted therapy, and CRC biology. However, several issues regarding clarity, experimental rigor, statistical robustness, and translational implications require careful consideration.
Questions:
- How do the authors reconcile the modest in vitro release percentages (≤8%) with the strong in vivo efficacy? Could accumulation over time explain this discrepancy?
- Did the authors measure serum or organ concentrations of VTD to verify targeted delivery?
- Are the effects observed in the xenograft and liver metastasis models statistically significant after correction for multiple testing?
- How might nanoparticle stability and release be affected under circulating serum proteases beyond MMP-7/9?
- Could the authors compare the performance of MSNs vs CCSMPs in the same in vivo model, rather than in separate experiments, for direct benchmarking?
- Have the authors evaluated long-term survival or recurrence in treated mice, beyond tumor size reduction?
- Is there any evidence that UBXN2A elevation is maintained after treatment cessation, or does it rapidly decline?
Author Response
We appreciate the reviewers for their time and valuable feedback. Their suggestions helped us improve this paper. We added several new results, including pharmacokinetic data on serum and organ accumulation of VTD, which confirms the anti-cancer activity and targeted delivery of MSN/COOH/VTD/CAS in our findings. These new results, along with improvements to the original results and the main text, address all the concerns raised by the reviewers. We are confident that we have answered all the questions point by point, in accordance with the reviewers’ comments. In the main manuscript, all edited or modified sections are traceable using the Word Track Changes feature.
Reviewer 1
The manuscript presents a novel nanoparticle-based strategy for targeted delivery of veratridine (VTD) to metastatic colorectal cancer (CRC), exploiting the overexpression of MMP-7/MMP-9 in tumor microenvironments. The authors demonstrate that casein-coated mesoporous silica nanoparticles (MSNs) and calcium carbonate submicroparticles (CCSMPs) enable MMP-responsive VTD release, leading to significant anti-tumor effects in both xenograft and spleno-hepatic metastasis models. The study also explores the mechanistic role of UBXN2A in mediating VTD-induced apoptosis.
The work is innovative and clinically relevant, combining nanomedicine, targeted therapy, and CRC biology. However, several issues regarding clarity, experimental rigor, statistical robustness, and translational implications require careful consideration.
Questions:
- How do the authors reconcile the modest in vitro release percentages (≤8%) with the strong in vivo efficacy? Could accumulation over time explain this discrepancy?
As reviewer #1 highlighted, the release rate is around 8%. However, the high surface area and tunable pore size of NPs ensure a more robust MMPs-casein interaction, resulting in a higher local concentration of VTD at the tumor site, providing a significant therapeutic impact on cancer cells [1]. More importantly, particles size of NPs (70-110 nm), which is ideal for the EPR effect, accumulates VTD inside the tumor over time and ensures an effective controlled release [2]. Finally, VTD can be confined in the ordered mesopores of NPs and sealed by casein coatings, which provides sustained VTD release kinetics [3]. We found several similar examples in the literature for nanoparticle and their slow-release mechanism. Hasseini et. al. showed that liposomes release 18% of the drug in vitro over 120 h (5 days) while in vivo PK/PD studies in an animal inflammation model achieved good therapeutic efficacy [4]. Although this is not directly an enzyme-triggered MSN nano-cargo, it is a similar slow-release drug carrier. Echeverri et al. found that the in vitro release of the drug carbamazepine (in coarse and nanometric emulsions) measured by dialysis was 2%, whereas in vivo adsorption was higher (20%) for oral administration in rabbits. Most importantly, nanosized droplets significantly increased the rate of adsorption [5]. This, too, is not an enzyme-responsive release; the test involves emulsion formation, which refers to passive release, and the efficacy is not expressed as a therapeutic outcome, but rather it’s the rate of adsorption. The above examples demonstrate that a slow-release carrier, through accumulation and sustained exposure, can compensate for low in vitro release and achieve therapeutic effects in vivo.
- Wijewantha, N., Eikanger, M. M., Antony, R. M., Potts, R. A., Rezvani, K., & Sereda, G. (2021). Targeting colon cancer cells with enzyme-triggered casein-gated release of cargo from mesoporous silica-based nanoparticles. Bioconjugate chemistry, 32(11), 2353-2365.
- Dogra, P., Adolphi, N. L., Wang, Z., Lin, Y. S., Butler, K. S., Durfee, P. N., ... & Brinker, C. J. (2018). Establishing the effects of mesoporous silica nanoparticle properties on in vivo disposition using imaging-based pharmacokinetics. Nature Communications, 9(1), 4551.
- Liu, Y., Zhao, M., Zhang, M., Yang, B., Qi, Y. K., & Fu, Q. (2025). Mesoporous Silica Nanoparticle-Based Nanomedicine: Preparation, Functional Modification, and Theranostic Applications. Materials Today Bio, 102223.
- Hosseini, S. H., Maleki, A., Eshraghi, H. R., & Hamidi, M. (2016). Preparation and in vitro/pharmacokinetic/pharmacodynamic evaluation of a slow-release nano-liposomal form of prednisolone. Drug Delivery, 23(8), 3008-3016.
- Echeverri, J. D., Alhajj, M. J., Montero, N., Yarce, C. J., Barrera-Ocampo, A., & Salamanca, C. H. (2020). Study of in vitro and in vivo carbamazepine release from coarse and nanometric pharmaceutical emulsions obtained via ultra-high-pressure homogenization. Pharmaceuticals, 13(4), 53.
- Did the authors measure serum or organ concentrations of VTD to verify targeted delivery?
In collaboration with Dr. Joseph Rower at the University of Utah's Center for Human Toxicology (CHT), we measured the concentration of Veratridine molecules in the serum and in tissues such as tumor, heart, liver, and brain. Supplementary Figure 8 displays the concentration of VTD (pg/mg) in tissues, as determined by mass spectrometry. These new results confirm that NP-VTD can effectively reduce VTD in non-cancerous organs while accumulating in the liver tissue, where metastatic tumors form. We observed improved performance of NP-VTD, especially in male mice. More notably, there was no detectable veratridine in the plasma of treated mice.
- Are the effects observed in the xenograft and liver metastasis models statistically significant after correction for multiple testing?
The xenograft data in Figure 5 include two mice per group, each with two tumors—one on the left flank and one on the right flank. As a result, the measurements in panels B and D are based on four tumors per group. The one-way ANOVA reveals no significant differences among groups, although a trend suggests that tumor tissue is more sensitive to VTD and NP-VTD in male mice. No similar trend was observed in female mice. Therefore, we concluded that the small sample size likely has low statistical power, making it difficult to detect any impact of VTD on tumor sizes if it exists. Based on these xenograft findings and the observed trend, we proceeded to a splenic-hepatic CRC mouse model, which more accurately mimics metastatic CRC. Figure 6 shows that MSN-COOH/VTD/CAS (VTD, 0.2 mg/kg) significantly reduced tumor size and number. As previously reported by our group, VTD has a more effective influence on tumor growth in male mice. The statistical analysis for the xenograft and liver metastasis mouse models in the main text was clarified to address the reviewer’s questions, using the insights provided in this section. Our aim was to demonstrate the feasibility of the MSN-COOH/VTD/CAS nanoparticles and their ability to reduce tumor growth in both the xenograft and splenic-hepatic mouse models of CRC. Initially, we planned to repeat both experiments with four mice, or eight tumors, per treatment group. After seeing the improved release strategy of the calcium nanoparticles (Figure 8), we decided to repeat these animal studies using the calcium nanoparticles in an ongoing project. This way, in our next manuscript, we will be able to compare the efficacy of the MSNs and the calcium nanoparticles in an animal model in both male and female mice.
- How might nanoparticle stability and release be affected under circulating serum proteases beyond MMP-7/9?
We agree with reviewer #1 that other MMPs should affect the particle’s stability, and they may trigger VTD release. Based on evidence in the literature, MMP-7 is substantially expressed by CRC; [1, 2] and MMP-7 concentration increases in a stage-dependent manner in CRC, which enables the NPs-VTD to efficiently target higher-stage metastatic tumors [3]. In addition, the activated form of MMP-7 is exclusively present in tumor masses, while it is absent in normal tissues [2]. Meanwhile, in several solid tumors, the level of MMP-7 within tumor tissues is approximately 6-fold greater than in normal cells [4]. This evidence strongly supports the absence of VTD in plasma in animals treated with pure VTD or NP-VTD. While exploring the effect of all known MMPs on NP-VTD is not practically justified, the in vivo efficiency of VTD speaks to the low interference of other MMPs on the therapeutic action of NP-VTD.
- Could the authors compare the performance of MSNs vs CCSMPs in the same in vivo model, rather than in separate experiments, for direct benchmarking
Comparing the performance of MSNs vs CCSMPs in the same in vivo model is one of our future objectives in this drug-delivery project. The results we present here support the proposed experiments, which are currently in progress and will be reported later.
- Have the authors evaluated long-term survival or recurrence in treated mice, beyond tumor size reduction?
We appreciate Reviewer #1 sharing this important question. To assess long-term survival or recurrence, we will use an APC mouse model that develops tumors in the descending colon and rectum, closely mimicking human CRC tumors. Additionally, we have an APC-based genetic model that reflects our VTD pharmacological model. We will complete and report the survival and recurrence experiments in our APC models in a future publication.
- Is there any evidence that UBXN2A elevation is maintained after treatment cessation, or does it rapidly decline?
We collected tumor tissues from the experiments shown in Figures 5 and 6. We are currently collaborating with a histology core at the University of Nebraska Medical Center to evaluate UBXN2A expression in liver tissues with tumors, shown in Figure 6. The histological analysis of UBXN2A, combined with the examination of downstream protein targets of UBXN2A with and without VTD treatments, will provide a clearer understanding of VTD's therapeutic effects on tumor tissues and adjacent normal tissues in the liver. As requested by reviewer #1, we employed Western blot (WB) experiments to measure UBXN2A levels in xenograft tumors shown in Figure 5. The WB results (n=2 per treatment) showed a significant increase in UBXN2A level in NP-VTD-treated mice compared to NP-empty. We did not observe a similar pattern in xenograft tumors from pure VTD-treated mice, likely due to the dominance of necrotic tissue within the tumors that developed during the 4-week treatment, as previously reported [5].
Reviewer 2 Report
Comments and Suggestions for Authors
The study titled “Liver-specific nanoparticle-mediated delivery and MMP-triggered release of Veratridine, a potent pro-apoptotic molecule, to effectively target metastatic colorectal cancer”. This study demonstrates that veratridine (VTD), delivered via MMP-responsive, casein-coated nanoparticles, selectively induces apoptosis in metastatic colorectal cancer cells and reduces tumor burden in mouse models, with calcium carbonate-based particles further enhancing targeted drug release and therapeutic efficacy while minimizing effects on healthy tissue; overall, the work presents a promising tumor-specific delivery strategy with strong translational potential, though further studies on long-term safety and human applicability are warranted.
- The abstract is very detailed and includes extensive methodological descriptions. Consider reducing it by focusing on the key findings, main therapeutic strategy, and primary outcomes, while omitting detailed experimental procedures and secondary comparisons.
- Does the introduction clearly justify the need for targeted therapies in metastatic colorectal cancer and the selection of VTD as a therapeutic agent?
- Is the explanation of the MMP-triggered nanoparticle delivery system concise and logically connected to the therapeutic hypothesis, or could it be streamlined for clarity?
- Do the results convincingly demonstrate that VTD activates both intrinsic and extrinsic apoptotic pathways across multiple CRC cell lines, and is the UBXN2A-dependency clearly supported?
- Does the AOM/DSS mouse model data effectively show that VTD suppresses tumor progression in a UBXN2A-dependent manner, and are the experimental design and endpoints sufficiently clear for interpretation?
- Consider including additional in vivo biodistribution or pharmacokinetic data to strengthen the claim of tumor-selective release.
- It may be helpful to clarify the rationale for the selected sample sizes and include statistical analysis to better support gender-specific observations.
- It would strengthen the study to perform in vivo experiments under tumor-mimicking acidic conditions, measuring cumulative VTD release and therapeutic efficacy, which could demonstrate the advantage of dual-trigger CCSMPs over MSNs for metastatic CRC treatment.
- According to the discussion, it significantly inhibits tumor progression while applying a nanosystem for therapeutics. In this regard, relevant studies on promoting tumor therapy with multiple interactions could be cited, such as ACS Nano, 2025,19, 2, 2117–2135; ACS Nano 2025, 19, 27, 25134; Nano Today, 2025, 65, 102838.
- Consider including a comparative in vivo pharmacokinetic study showing VTD concentrations at the tumor site versus normal organs to confirm that the faster release of CCSMPs translates to enhanced tumor targeting without increasing systemic exposure.
Author Response
We appreciate the reviewers for their time and valuable feedback. Their suggestions helped us improve this paper. We added several new results, including pharmacokinetic data on serum and organ accumulation of VTD, which confirms the anti-cancer activity and targeted delivery of MSN/COOH/VTD/CAS in our findings. These new results, along with improvements to the original results and the main text, address all the concerns raised by the reviewers. We are confident that we have answered all the questions point by point, in accordance with the reviewers’ comments. In the main manuscript, all edited or modified sections are traceable using the Word Track Changes feature.
Reviewer 2
The study titled “Liver-specific nanoparticle-mediated delivery and MMP-triggered release of Veratridine, a potent pro-apoptotic molecule, to effectively target metastatic colorectal cancer”. This study demonstrates that veratridine (VTD), delivered via MMP-responsive, casein-coated nanoparticles, selectively induces apoptosis in metastatic colorectal cancer cells and reduces tumor burden in mouse models, with calcium carbonate-based particles further enhancing targeted drug release and therapeutic efficacy while minimizing effects on healthy tissue; overall, the work presents a promising tumor-specific delivery strategy with strong translational potential, though further studies on long-term safety and human applicability are warranted.
- The abstract is very detailed and includes extensive methodological descriptions. Consider reducing it by focusing on the key findings, main therapeutic strategy, and primary outcomes, while omitting detailed experimental procedures and secondary comparisons.
We agreed with Reviewer #2 and improved the abstract to highlight the key points.
- Does the introduction clearly justify the need for targeted therapies in metastatic colorectal cancer and the selection of VTD as a therapeutic agent?
We agreed with Reviewer #2 and added a paragraph highlighting the current challenge with metastatic CRC, including low survival rates, a high rate of drug resistance, and limited effective therapies.
- Is the explanation of the MMP-triggered nanoparticle delivery system concise and logically connected to the therapeutic hypothesis, or could it be streamlined for clarity?
We improved the main text to further clarify the causal connection between the MMP-triggered nanoparticle delivery system and the therapeutic hypothesis proposed in this study.
- Do the results convincingly demonstrate that VTD activates both intrinsic and extrinsic apoptotic pathways across multiple CRC cell lines, and is the UBXN2A-dependency clearly supported?
Based on previous publications, the underlying mechanism for the intrinsic pathway mainly involves the suppressive effect of the VTD-UBXN2A axis on mitochondrial heat shock protein (mortalin). We decided to examine the extrinsic pathway, as there is strong evidence from the literature that natural anticancer therapeutics simultaneously target both the intrinsic and extrinsic pathways of apoptosis. Several mechanisms have been reported for those plant-based extracts with both intrinsic and extrinsic pathways [1].
There are three potential mechanisms that enable VTD to activate the extrinsic pathway: a. VTD can function as a ligand for death receptors, b. cytoplasmic VTD can transcriptionally increase the levels of death receptors or components of the extrinsic pathway, c. VTD can affect the levels of extracellular MMPs, thereby promoting the activities of death receptors, as previously described [2].
- Does the AOM/DSS mouse model data effectively show that VTD suppresses tumor progression in a UBXN2A-dependent manner, and are the experimental design and endpoints sufficiently clear for interpretation?
The AOM/DSS model showed that the presence of VTD significantly decreases tumor growth in WT mice that received 0.1mg/kg VTD compared to WT mice that received 0.1 % ethanol (vehicle). More importantly, the results in Figure 2 indicate that the UBXN2A heterozygous mouse (UBXN2A +/-) also responded to VTD, although the effect was less strong in terms of tumor growth. This is likely due to the half-level expression of UBXN2A in haplosufficient UBXN2A mice [3]. The UBXN2A +/- mouse model in this study mimics 50% population of CRC patients who have low expression of UBXN2A [4] and therefore can benefit from VTD by enhancing the expression of UBXN2A through the transcriptional activities pathway regulated by VTD, as a plant alkaloid.
- Consider including additional in vivo biodistribution or pharmacokinetic data to strengthen the claim of tumor-selective release.
We have completed a set of PK studies in the xenograft CRC tumors and added them as a new figure to the supplementary data (Figure S8). We included a statement in the main text to address the questions above. In addition, We are currently developing a CRC APC mouse model that develops tumors in the descending colon and rectum, mimicking human CRC. This APC model is a more physiologically relevant model for CRC, allowing us to evaluate safety, therapeutic efficacy, and conduct comprehensive PK/PD studies. The results obtained from the APC mouse model and its associated PK/PD studies will be reported in the next manuscript.
- It may be helpful to clarify the rationale for the selected sample sizes and include statistical analysis to better support gender-specific observations.
The xenograft data in Figure 5 involve two mice per group, with each mouse having two tumors implanted on either flank. As a result, the measurements in panels B and D are based on four tumors per group. The one-way ANOVA shows no significant differences among groups, although there is a trend indicating tumor tissue sensitivity to pure VTD and NP-VTD in male mice. No similar trend was observed in female mice. Therefore, we conclude that the small sample size likely has low statistical power, making it difficult to detect any effect of VTD on tumor sizes if one exists. Based on these xenograft findings and the observed trend, we moved to a splenic-hepatic CRC mouse model, which more closely mimics metastatic CRC. Figure 6 demonstrates that MSN-COOH/VTD/CAS (VTD, 0.2 mg/kg) significantly reduced tumor size and number compared to NP-empty treatment. The statistical analysis for the xenograft and liver metastasis mouse models in the main text was revised to address the reviewer’s questions, incorporating the insights provided in this section.
- It would strengthen the study to perform in vivo experiments under tumor-mimicking acidic conditions, measuring cumulative VTD release and therapeutic efficacy, which could demonstrate the advantage of dual-trigger CCSMPs over MSNs for metastatic CRC treatment.
We agreed with the reviewer #2's suggestion, since tumors create an acidic microenvironment. The in vivo experiments with calcium-based nanoparticles will serve as an ideal model, as the CCSMP release rate can be controlled by the acidic environment of metastatic tumors. The release mechanism in the MSN-based nanoparticle used in this study responds more effectively to the proteolytic activity of MMPs than to the acidity of the tumors.
- According to the discussion, it significantly inhibits tumor progression while applying a nanosystem for therapeutics. In this regard, relevant studies on promoting tumor therapy with multiple interactions could be cited, such as ACS Nano, 2025,19, 2, 2117–2135; ACS Nano 2025, 19, 27, 25134; Nano Today, 2025, 65, 102838.
Thank you for sharing this relevant reference. We added these to the discussion section when we were discussing potential combination therapy (Reference 60).
- Consider including a comparative in vivo pharmacokinetic study showing VTD concentrations at the tumor site versus normal organs to confirm that the faster release of CCSMPs translates to enhanced tumor targeting without increasing systemic exposure.
Definitely. The results we present here validate the proposed experiments, which are currently in progress and will be reported in a later publication.
- Rajabi, S., et al., The Most Competent Plant-Derived Natural Products for Targeting Apoptosis in Cancer Therapy. Biomolecules, 2021. 11(4).
- Fu, H., et al., Matrix metalloproteinase-7 protects against acute kidney injury by priming renal tubules for survival and regeneration. Kidney Int, 2019. 95(5): p. 1167-1180.
- Sane, S., et al., UBXN2A enhances CHIP-mediated proteasomal degradation of oncoprotein mortalin-2 in cancer cells. Mol Oncol, 2018.
- Abdullah, A., et al., A plant alkaloid, veratridine, potentiates cancer chemosensitivity by UBXN2A-dependent inhibition of an oncoprotein, mortalin-2. Oncotarget, 2015. 16(27): p. 23561-81.
Reviewer 3 Report
Comments and Suggestions for Authors
In this study, the authors hypothesized that targeted elevation of UBXN2A in cancer cells through site-specific delivery of VTD inhibits tumor development. The proven anti-metastatic activity of VTD combined with MSN-based drug delivery makes it a promising new targeted therapy for patients with metastatic CRC. The results reported in this paper show that the excessive release of MMP-7 and MMP-9 present in metastatic colon cancer cells triggers the release of VTD from casein-coated MSNs and calcium carbonate micro- and submicro-particles. This study is interesting especially broaden the material platform for developing targeted, gated delivery of anti-cancer therapeutics.
I have a few comments as follows:
- What kind of methods that the authors were employed to validate the delivery efficiency of nanoparticles?
- Follow-up the last question, any evidence to validate its safety, any side effects were observed in the pre-clinical mouse studies?
- Is there any correlation between treatment response and genetic mutation background in this experimental setting?
- The molecular weight of internal control in Figure 1A is missing.
- One more dosage of VTD for example at 30μM is highly recommend to incorporate into the western blot(Figure 1A&D).
Author Response
We appreciate the reviewers for their time and valuable feedback. Their suggestions helped us improve this paper. We added several new results, including pharmacokinetic data on serum and organ accumulation of VTD, which confirms the anti-cancer activity and targeted delivery of MSN/COOH/VTD/CAS in our findings. These new results, along with improvements to the original results and the main text, address all the concerns raised by the reviewers. We are confident that we have answered all the questions point by point, in accordance with the reviewers’ comments. In the main manuscript, all edited or modified sections are traceable using the Word Track Changes feature.
Reviewer 3
In this study, the authors hypothesized that targeted elevation of UBXN2A in cancer cells through site-specific delivery of VTD inhibits tumor development. The proven anti-metastatic activity of VTD combined with MSN-based drug delivery makes it a promising new targeted therapy for patients with metastatic CRC. The results reported in this paper show that the excessive release of MMP-7 and MMP-9 present in metastatic colon cancer cells triggers the release of VTD from casein-coated MSNs and calcium carbonate micro- and submicro-particles. This study is interesting especially broaden the material platform for developing targeted, gated delivery of anti-cancer therapeutics.
I have a few comments as follows:
- What kind of methods that the authors were employed to validate the delivery efficiency of nanoparticles?
In this study, the significant response of liver tumors to NP-VTD demonstrated how tumor tissues react to the delivered VTD in the liver, where metastatic tumors developed. We will complete the IHC and WB as well as mass spectrometry experiments on the collected tumors and their adjacent normal tissue, as demonstrated in Figure 6. Measurement of VTD concentration, as well as its impact on UBXN2A expression and its downstream protein targets, will be reported in future publications.
- Follow-up the last question, any evidence to validate its safety, any side effects were observed in the pre-clinical mouse studies?
We have already established the safety, therapeutic effectiveness, and VTD accumulation of pure VTD in a pre-clinical mouse model [1]. Please also see the answer provided for the previous question.
- Is there any correlation between treatment response and genetic mutation background in this experimental setting?
Thank you for the excellent suggestion. In our previous publication, we found that the endogenous level of UBXN2A is elevated in the early stage of CRC (well-differentiated tumors) and significantly decreases in the late stage of CRC (poorly differentiated tumors). We concluded that the stage-dependent upregulation and downregulation of UBXN2A in tumors may be a response to the typical order of mutations that develops in CRC, as observed in other tumor suppressor proteins. We are currently examining the response of a set of patient-derived xenograft (PDX) cell lines with diverse mutation profiles to pure VTD and NP-VTD using the 3D model. Completion of the PDX cell lines study may generate a correlation between the genetic mutation background of CRC tumors and the therapeutic efficacy of VTD treatment.
- The molecular weight of internal control in Figure 1A is missing.
We fixed this issue in Figure 1.
- One more dosage of VTD for example at 30μM is highly recommend to incorporate into the western blot(Figure 1A&D).
Our previously published results have shown that pure VTD induces apoptosis and decreases cell proliferation in a dose-dependent manner. We have previously published data after 10μM, 30μM, 100μM, 200μM, and 300μM [2]. We found that 100μM is the IC50 of VTD for inhibiting cell growth by 50% using the MTT assay. Therefore, we did not repeat the WB experiment for the level of UBXN2A in panels F and H in Figure 1.
- Freeling, J.L., et al., Pre-clinical safety and therapeutic efficacy of a plant-based alkaloid in a human colon cancer xenograft model. Cell Death Discov, 2022. 8(1): p. 135.
- Abdullah, A., et al., A plant alkaloid, veratridine, potentiates cancer chemosensitivity by UBXN2A-dependent inhibition of an oncoprotein, mortalin-2. Oncotarget, 2015. 16(27): p. 23561-81.
Reviewer 4 Report
Comments and Suggestions for Authors
Dear authors,
Here are my comments on the article “Liver-specific nanoparticle-mediated delivery and MMP-triggered release of Veratridine, a potent pro-apoptotic molecule, to effectively target metastatic colorectal cancer”.
- I recommend that authors remove “a potent pro-apoptotic molecule” from the title. The first letter of each word should be capitalized.
- I suggest removing the simple summary. The abstract seems long, so please make it concise.
- The introduction started with an abbreviation (CRC); please provide its full name. The full name of the abbreviations where it is first used should be provided, although the abbreviation terms used in the articles are listed on page 26.
- Include only 5 keywords
- Scheme 1 should go to the materials and methods section.
- Please check the line spacing in the article throughout.
- Please mention the concentration of sodium hydroxide used in the MSN synthesis and surface functionalization. What was the initial temperature before it was raised to 80°C?
- Under what conditions was the calcination performed, in air or nitrogen? Do you have a picture of the sample before (the particles dried at 60°C for 12h) and after the calcination at 500°C for 5 h (final product MCM-41 MSN)?
- Please perform the FTIR of the pristine and functionalized mesoporous silica nanoparticle to confirm the presence of -NH2 and -COOH groups on the surface. Indicates the peaks in the FTIR spectrum. Simply providing the zeta potential values does not confirm the presence of these functional groups. Zeta potential measures changes in charges/potential surrounding ions.
- Is there a change in the morphology of the MSN after surface functionalization and the size of the particles? How was the measurement in size taken? In addition to this, I suggest that authors provide more SEM images. Figure 3 can be separated into three, so that we have a clear visualization/representation of the SEM images and graphs.
My recommendation is to address the above comments and resubmit the manuscript.
Author Response
Here are my comments on the article “Liver-specific nanoparticle-mediated delivery and MMP-triggered release of Veratridine, a potent pro-apoptotic molecule, to effectively target metastatic colorectal cancer”.
We appreciate Reviewer #4 for the time and valuable feedback. The suggestions helped us improve this paper. We added several new results, including pharmacokinetic data on serum and organ accumulation of VTD, which confirms the anti-cancer activity and targeted delivery of MSN/COOH/VTD/CAS in our findings. These new results, along with improvements to the original results and the main text, address all the concerns raised by the reviewers, including reviewer #4. We are confident that we have answered all the questions point by point, in accordance with the reviewers’ comments. In the main manuscript, all edited or modified sections are traceable using the Word Track Changes feature.
1.
I recommend that authors remove “a potent pro-apoptotic molecule” from the title. The first letter of each word should be capitalized.
We agree with Reviewer #4, and we revised the title accordingly.
2.
I suggest removing the simple summary. The abstract seems long, so please make it concise.
We kept the “simple summary section” in the paper as we followed the journal's format for preparing a research article. According to the instructions for authors, a research article should contain a simple summary section.
3.
The introduction started with an abbreviation (CRC); please provide its full name. The full name of the abbreviations where it is first used should be provided, although the abbreviation terms used in the articles are listed on page 26.
Thanks for the note. We corrected the first sentence accordingly.
4.
Include only 5 keywords
The revised version of the main text has 5 keywords.
5.
Scheme 1 should go to the materials and methods section.
We agree with reviewer #4, and we have converted Scheme 1 into a Graphic Abstract, serving as a visual complement to the written introduction section.
6.
Please check the line spacing in the article throughout.
We rechecked the entire article to ensure it complies with the Journal of Cancers format, including spacing (justified, at least 12) and font (Palatino Linotype).
- Please mention the concentration of sodium hydroxide used in the MSN synthesis and surface functionalization. What was the initial temperature before it was raised to 80°C?
Ans: Thank you for pointing this out. We have added the concentration of sodium hydroxide (2.00 M) used in the MSN synthesis. The initial temperature before it was raised to 80°C was room temperature.
- Under what conditions was the calcination performed, in air or nitrogen? Do you have a picture of the sample before (the particles dried at 60°C for 12h) and after the calcination at 500°C for 5 h (final product MCM-41 MSN)?
Ans: Thank you for pointing this out. The calcination was done at 550°C in air for 5 h, and we have added the condition ‘in air’ to the manuscript. There was no significant change in appearance for before and after calcination samples, which is consistent with our published SEM and TEM images (reference 32).
- Please perform the FTIR of the pristine and functionalized mesoporous silica nanoparticle to confirm the presence of -NH2 and -COOH groups on the surface. Indicates the peaks in the FTIR spectrum. Simply providing the zeta potential values does not confirm the presence of these functional groups. Zeta potential measures changes in charges/potential surrounding ions.
Ans: The FTIR data for MSN, MSN-NH2, and MSN-COOH are already published in our group’s recent paper (Figure 2C in the reference no. 32 in this manuscript reproduced below for your convenience).
Reference 32: Figure 2. (C) Fourier transform infrared spectroscopy (FTIR) spectra of (a) MSNs, (b) MSNs-NH2, and (c) MSNs-COOH, and (d) MSNs/ HAPs.
- Is there a change in the morphology of the MSN after surface functionalization and the size of the particles? How was the measurement in size taken? In addition to this, I suggest that authors provide more SEM images. Figure 3 can be separated into three, so that we have a clear visualization/representation of the SEM images and graphs.
Ans: There is no change in morphology; pristine and functionalized particles are spherical in shape. After functionalization, the particle size increases by approximately 10 to 15 nm. The size measurements were obtained from SEM size distribution data and hydrodynamic diameter data from DLS measurement. All these data are published in our group’s last paper (Reference no. 32 in this manuscript). Please refer to Figure S1(a-b) and Figure S2(a-f) of Reference 32—reproduced below for your convenience.
Reference 32: Figure S1. SEM micrographs of A) MSNs-NH2, (B) MSNs-COOH, Aminated and carboxylated MSNs show almost spherical shape morphology where it suggests that surface modification to MSNs didn’t affect the shape of the particles.
Reference 32: Figure S2. Size distribution of a) MSNs, b) MSNs-NH2, c) MSNs-COOH and DLS measurements to determine the hydrodynamic diameter of d) MSNs, e) MSNs-NH2, f) MSNs-COOH
Round 2
Reviewer 2 Report
Comments and Suggestions for Authors
All my concerns have been addressed. The manuscript is now acceptable for publication in its present form.
Author Response
We appreciate Reviewer #2 for all valuable comments, which improved the clarity of the presented data in this manuscript.
Reviewer 4 Report
Comments and Suggestions for Authors
In the revised version of the manuscript, the authors have addressed all my comments. I suggest the authors round the volume of the TEOS used (11.477 mL) to the nearest hundredth value (11.47 mL). (Please see section 2.3; MSN Synthesis and Surface Functionalization; line 189). For comments 8, 9, and 10, the authors have not provided the pictures but have given explanations based on their previously published data on characterization and functionalization (reference 32). I am ok with the author’s responses to these comments. Lastly, I suggest the author check the title in the supplementary information and ensure it matches the main.
Author Response
We appreciate Reviewer #4 for the final points, which we fixed in the main text and supplementary data, respectively.